# *In vitro* activity of AST-120 that suppresses indole signaling in *Escherichia coli*, which attenuates drug tolerance and virulence

**Hidetada Hirakawa**[1]*, **Motoyuki Uchida**[2], **Kumiko Kurabayashi**[1], **Fuyuhiko Nishijima**[2], **Ayako Takita**[1], **Haruyoshi Tomita**[1,3]

**1** Department of Bacteriology, Gunma University, Graduate School of Medicine, Maebashi, Gunma, Japan,
**2** Pharmaceuticals and Agrochemicals Division, Kureha Corporation, Shinjuku-ku, Tokyo, Japan,
**3** Laboratory of Bacterial Drug Resistance, Gunma University, Graduate School of Medicine, Maebashi, Gunma, Japan

* hirakawa@gunma-u.ac.jp

**Data Availability Statement:** All relevant data are within the manuscript.

## Abstract

AST-120 (Kremezin) is used to treat progressive chronic kidney disease (CKD) by adsorbing uremic toxin precursors produced by gut microbiota, such as indole and phenols. In this study, we propose that AST-120 reduces indole level, consequently suppresses indole effects on induction of drug tolerance and virulence in *Escherichia coli* including enterohaemorrhagic strains. In experiments, AST-120 adsorbed both indole and tryptophan, a precursor of indole production, and led to decreased expression of *acrD* and *mdtEF* which encode drug efflux pumps, and elevated *glpT*, which encodes a transporter for fosfomycin uptake and increases susceptibility to aztreonam, rhodamine 6G, and fosfomycin. AST-120 also decreased the production of EspB, which contributes to pathogenicity of enterohaemorrhagic *E. coli* (EHEC). Aztreonam, ciprofloxacin, minocycline, trimethoprim, and sulfamethoxazole were also adsorbed by AST-120. However, fosfomycin, in addition to rifampicin, colistin and amikacin were not adsorbed, thus AST-120 can be used together with these drugs for therapy to treat infections. These results suggest another benefit of AST-120, i.e., that it assists antibacterial chemotherapy.

## Introduction

The spread of pathogens that have developed antimicrobial resistance (AMR), and the shortage of new antibiotics are critical issues in health care. Alternatives to traditional bacteriostatic and bactericidal agents have been proposed, such as reagents targeting bacterial proteins responsible for pathogenicity and/or drug resistance, including antitoxin agents and extended-spectrum β-lactamase (ESBL) inhibitors, respectively [1–2]. Such reagents, which assist therapies by using traditional drugs to treat infections, are associated with low rates of resistance, presumably because they do not affect bacterial growth.

Drug resistance is characterized as the inherited ability of bacteria to grow in a concentration of a drug that would otherwise prevent growth, and it is typically acquired by genetic

**Funding:** The authors, Dr. Uchida and Dr. Nishijima belong to a commercial company, Kureha corp. They contributed to the study design, data interpretation and providing the AST-120 material in addition to financial support (20,000 dollars) to perform experiments and publish the results, but not salaries for authors as the funder. We believe that the financial support should be reasonable value, and I suggest that they did not directly participate in data collection. HH was supported by JSPS KAKENHI "Grant-in-Aid for Scientific Research (C) Grant Number 19K07533 (https://www.jsps.go.jp/) and HT was supported by Japan Agency for Medical Research and Development, AMED Grant Number 19fk0108061h0502 (https://www.amed.go.jp/en/). The funders had no role in study design, data collection and analysis, decision to publish, or preparation of the manuscript.

**Competing interests:** The authors, Dr. Uchida and Dr. Nishijima belong to a commercial company, Kureha corp. This commercial affiliation does not alter our adherence to PLOS ONE policies on sharing data and materials.

modifications. In addition to drug resistance, drug tolerance of bacteria is recently re-considered as a potential critical issue. Drug tolerance is defined as a transient ability of bacteria to survive under bactericidal antibiotic treatments without having acquired a resistance phenotype, therefore, it is associated with the failure of antimicrobial chemotherapy [3–4].

Bacteria use particular signaling molecules, such as autoinducers, when they contact other bacteria, then alter the expression profile of subsets of genes [5]. In most of bacterial species, the communication mediated by these signaling compounds is closely associated with drug resistance, drug tolerance and pathogenicity, making them candidate targets for the treatment of refractory infectious diseases caused by pathogens resistant to commonly used antibiotics [6].

Bacterial signaling molecules have been the subject of extensive studies, particularly in pathogens. Many species of gram-negative bacteria use acyl-homoserine lactones and a derivative of 4,5-dihydroxy-2,3-pentanedione (DPD) termed autoinducer-2 [7–8]. As an exception, *Escherichia coli* strains do not produce acyl-homoserine lactones, but they produce autoinducer-2. In addition to autoinducer-2, *E. coli* also utilizes indole [9]. Tryptophanase (TnaA) is an enzyme that catalyzes a reaction from tryptophan into pyruvate, then indole is generated as a by-product.

Indole controls diverse cellular functions of bacteria including drug resistance, drug tolerance, biofilm formation and virulence as a signal molecule [10–12]. Previously, we found that indole increases expression of genes that encode proteins for drug efflux pumps and the type III secretion system, which contribute to drug tolerance and pathogenicity in host epithelial cells, respectively, in species, such as the food-borne pathogen enterohaemorrhagic *E. coli* (EHEC) [13–14]. On the other hand, indole decreases expression of *glpT*, which encodes the uptake transporter of fosfomycin with glycerol-3-phospate, leading to increased tolerance to fosfomycin [15].

In human host, indole is produced by some of enteric bacteria including *E. coli*. That is absorbed into the intestine, and transported by the blood to the liver, where it is converted into indoxyl sulfate, and reenters the bloodstream [16]. Indoxyl sulfate is known to be a uremic toxin [17]. When kidney function is normal, indoxyl sulfate enters the renal tubular cells and is eventually excreted from the urine [18]. However, indoxyl sulfate remains highly in patients with chronic kidney disease (CKD), leading to an accelerated decline of kidney function [19].

AST-120 (Kremezin) is an oral carbonaceous adsorbent, used to treat progressive CKD by delaying the progression of the disease, followed by the initiation of dialysis [17, 20]. In enteric sites, AST-120 adsorbs uremic toxin precursors, including indole, produced by enteric bacteria, and reduces the production of uremic toxins, such as indoxyl sulfate [21].

We hypothesized that AST-120 adsorbs indole produced by *E. coli* and then attenuates indole signaling of *E. coli* species, including EHEC, which in turn decreases drug tolerance and type III secretion system-associated virulence elevated by indole. In this study, we assessed the *in vitro* role of AST-120 as a potential inhibitor of indole signaling. We show evidence that AST-120 can attenuate drug tolerance and virulence induced by indole signaling.

## Materials and methods

### Bacterial strains, plasmids and culture conditions

The bacterial strains and plasmids used in this study are described in Table 1. Bacteria were grown in LB (Luria-Bertani) medium or DMEM (Dulbecco's Modified Eagle Medium), and cell growth was monitored by absorbance at 600 nm. The pNNacrD-P [13], pNNgadE-P [13], and pNNglpT-P [22] are *lacZ* fusion reporter plasmids that monitor promoter activities of *acrD*, *gadE* and *glpT*, respectively. Each DNA fragment containing its promoter sequence was

**Table 1. Strains and plasmids used in this study.**

| Strain or plasmid | Relevant genotype/phenotype | Reference |
|---|---|---|
| Strains | | |
| MG1655 | *E. coli* K-12; wild type; reporter strain | [24] |
| MC4100 | *E. coli* K12, lacking the endogenous *lacZ* gene | [25] |
| MC4100ΔacrB | *acrB* mutant from MC4100 | [13] |
| O157:H7 Sakai | Wild-type parent EHEC O157:H7 (RIMD 0509952) | [26] |
| O157ΔtnaA | *tnaA* mutant from O157:H7 Sakai | [14] |
| Plasmids | | |
| pNNacrD-P[a] | *acrD* promoter reporter; Cm$^R$ | [13] |
| pNNgadE-P[a] | *gadE* promoter reporter; Cm$^R$ | [13] |
| pNNglpT-P | *glpT* promoter reporter; Cm$^R$ | [22] |

Cm$^R$: Chloramphenicol resistance

[a] Formally named pNNacrD and pNNyhiE, respectively

ligated to promoter-less *lacZ* on a pNN387 plasmid [23], allowing the promoter activity to be assessed according to β-galactosidase levels from LacZ expression on the plasmid. For the maintenance of plasmids, chloramphenicol was added to the growth media at a concentration of 30 μg/ml.

## EspB antiserum

EspB antiserum was produced in rabbits by Sigma-Aldrich Co. LLC (St. Louis, MO). A peptide synthesized from positions 294 to 312 of EspB was used as an antigen.

## Measurement of indole in bacterial culture

To determine indole levels in *E. coli* culture, we measured the indole concentration of the culture supernatant because most of indole produced by *E. coli* was secreted into culture media [27]. Bacteria were grown in the LB medium for 24 hours, and the cell cultures were centrifuged to remove the cell pellets. Indole in the culture supernatants was extracted with ethyl acetate, then 0.7 ml of Kovac's reagent (10 g of *p*-dimethylaminobenzaldehyde, 50 ml of HCl and 150 ml of amylalcohol) was added to 0.02 mL of the ethyl acetate fraction containing indole. The indole concentration was calculated by absorbance at 540 nm according to the standard curve using commercial indole (FUJIFILM Wako Pure Chemical Industries Corp., Osaka, Japan).

## Minimum inhibitory concentration (MIC) assays

MIC assays were performed by a serial agar dilution method. Bacteria were grown with or without indole in LB medium to the early-stationary phase. Five microliters of 100-fold-diluted cultures (~5,000 cells) was inoculated onto a LB agar plate containing drugs with or without indole and incubated for 16 h at 37°C. The MICs were determined as the lowest concentration at which growth was inhibited.

## Survival assays

Bacteria were grown with or without indole and/or AST-120 in LB medium to the mid-logarithmic phase, then 5.0 x 10$^7$ cells were transferred into the fresh LB medium containing either drug; 25 μg/ml aztreonam, 1.56 μg/ml fosfomycin or 75 μg/ml rhodamine 6G, and incubated

with shaking. At indicated time points, aliquots of the cultures were sampled, and the number of viable cells was determined as the colony forming unit (CFU) by plating serial dilutions on LB agar. Percentage survival was calculated as the value of CFU for cells after incubated with drug relative to the value of CFU for cells prior to incubation with drug as time 0.

### Promoter assays

Promoter activities were evaluated as β-galactosidase activities from LacZ expression on the reporter plasmids. The *E. coli* MC4100 strains carrying pNN-glpT-P, pNN-acrD-P or pNN-gadE-P, the LacZ reporter plasmids, were grown at 37°C in LB medium. β-Galactosidase levels from LacZ expression were determined using a Tropix Galacto-Light Plus according to the manufacturer's protocol (Thermo Fisher Scientific, Waltham, MA) as previously indicated [28].

### Measurement of tryptophan, aztreonam, ciprofloxacin, rhodamine 6G, rifampicin, minocycline, trimethoprim, sulfamethoxazole, fosfomycin and colistin in bacteria-free aqueous solution after treatment of AST-120

Each compound in aqueous solution was incubated for 2 hours in the presence and absence of 30 mg AST-120. Drug concentrations except fosfomycin, colistin and amikacin were calculated by measuring absorbance at 280 nm. Concentrations of fosfomycin and colistin and amikacin were measured by a diffusion disc assay using the *E. coli* MG1655 reporter strain as previously indicated [15] because these compounds have no aromatic ring structure, therefore do not absorb the light of 280 nm wavelength.

### EspB detection

EHEC strains were grown at 37°C with shaking to the early stationary phase in DMEM, and separated by centrifugation and filtration. Secreted proteins were precipitated from the supernatants with 10% trichloroacetic acid (TCA) and dissolved in Laemmli sample buffer (Bio-Rad Laboratories, Hercules, CA). The solution was boiled, and the proteins were separated on a 12.5% acrylamide Tris-glycine SDS/PAGE gel. The gel was electroblotted onto a polyvinylidene fluoride (PVDF) membrane (Bio-Rad Laboratories, Hercules, CA). EspB was detected with EspB antiserum with secondary anti-rabbit horseradish peroxidase-conjugated immunoglobulin G (IgG) (Sigma-Aldrich Co. LLC., St. Louis, MO) and SuperSignal West Pico Kit (Thermo Fisher Scientific, Waltham, MA). Protein bands of EspB were visualized on a LAS-4010 Luminescent image analyzer (GE Healthcare Japan, Tokyo).

## Results

### AST-120 adsorbs not only indole produced by *E. coli*, but also tryptophan, a substrate of indole production, which decreases the level of indole in media

To confirm the ability of AST-120 to adsorb indole, we incubated indole with AST-120 in a bacterial-free solution and then measured the level of indole in the supernatant after the removal of AST-120. *E. coli* strains, including EHEC produce indole in the range of 0.3–0.7 mM when grown in LB medium [27]. Indole is produced from tryptophan, then its production can be increased by exogenous addition of the tryptophan substrate. We found that *E. coli* strains, including the K-12 laboratory and EHEC strains MG1655 and O157:H7 Sakai, respectively, produced 2 mM indole when grown to the stationary phase in LB medium containing tryptophan, therefore we tested 2 mM indole to examine the effect of AST-120 in this study. More than 98% of the indole was adsorbed when incubated with at least 30 mg AST-120 (Fig

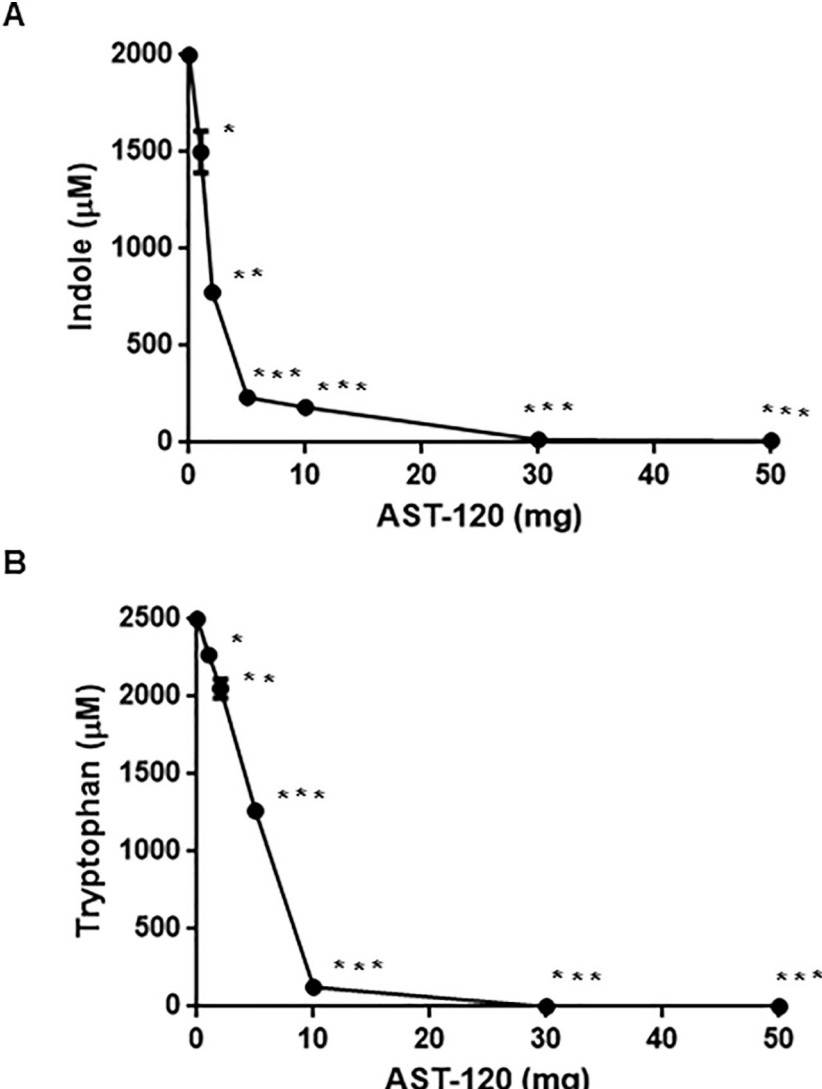

**Fig 1. Determination of indole and tryptophan levels.** (A) Indole concentration in supernatant from bacteria-free solution containing 2 mM indole after incubated for 2 hours with or without AST-120. (B) Tryptophan concentration in the supernatant from bacteria-free solution containing 2.5 mM tryptophan after incubated for 2 hours with or without AST-120. Data plotted are the means from three independent experiments; error bars indicate the standard deviations, *, $P<0.05$; **, $P<0.01$; ***, $P<0.001$. Asterisks denote significance for values of indole or tryptophan concentration in supernatant after incubated with AST-120 relative to those after incubated without AST-120.

1A). Indole has a bicyclic structure consisting of a benzene ring fused to a pyrrole ring. Because tryptophan has the same bicyclic structure, we tested whether AST-120 also adsorbs tryptophan. Our results showed that almost all 2.5 mM tryptophan in solution was adsorbed when incubated with 30 mg AST-120 (Fig 1B). These findings indicate that AST-120 adsorbs both indole and tryptophan, which leads to a reduction in indole levels in *E. coli* culture.

## AST-120 decreases tolerance to aztreonam, fosfomycin, and rhodamine 6G induced by indole

Indole induces expression of several genes that encode drug efflux pumps including *acrD* and *mdtEF*, and suppresses expression of *glpT*, which encodes a glycerol-3-phosphate/fosfomycin

symporter responsible for fosfomycin uptake [13, 15, 27]. AST-120 may minimize the effects of indole on expression of these genes, leading to an increase of drug susceptibility. To test this hypothesis, we measured promoter activities of *acrD*, *mdtEF*, and *glpT* in MC4100, an *E. coli* host that lacks a chromosomal *lacZ* gene when grown in the presence or absence of indole and AST-120. Because the *mdtEF* genes are transcribed with *gadE*, which is located upstream of *mdtEF*, we used the *gadE* promoter–*lacZ* reporter plasmid to assess expression of *mdtEF*. Consistent with our previous results, indole increased LacZ expression from *acrD* and *gadE* promoters while decreasing expression from the *glpT* promoter (Fig 2A). However, when grown in the presence of AST-120, no effect of indole on LacZ expression was observed (Fig 2A). *acrD* and *mdtEF* overexpression decrease susceptibility to aztreonam and rhodamine 6G, respectively while reduction of *glpT* expression decreases susceptibility to fosfomycin [15, 29–30]. We tested the susceptibilities of the MC4100 strain to aztreonam and fosfomycin when grown with or without indole and AST-120. To test the susceptibility to rhodamine 6G, we used MC4100ΔacrB as a host strain because *E. coli* is not susceptible to rhodamine 6G due to the AcrAB drug efflux pump, which is constitutively produced [30]. Consistent with previous studies, bacteria grown with indole were lower susceptible to these drugs than those grown without indole. In addition, we here showed that indole increases tolerance to these drugs. Drug tolerance is characterized by a slower killing even at high concentrations of the drug while MIC of the drug is not changed in contrast to drug resistance [3]. Indole addition did not increase MICs of aztreonam, fosfomycin and rodamine 6G in cultures of the MC4100 and MC41000ΔacrB strains (data not shown). On the other hand, compared to the strains grown without indole, the strains grown with indole were slowly killed during exposure of these drugs, suggesting that indole increases tolerance, but not resistance, (Fig 3A–3C). When AST-120 was present, the strains grown with indole were similarly killed to those grown without indole (Fig 3A–3C). We also note that AST-120 addition does not affect innate tolerance to aztreonam and fosfomycin because these killing curves for cells grown without indole in the presence of AST-120 were similar to those in the absence of AST-120. On the other hand, AST-120 addition moderately prolonged bacterial survival for cells grown without indole during exposure of rhodamine 6G although we do not know the reason. As a consequence, AST-120 suppresses the effect of indole on induction of AcrD, MdtEF, and GlpT-associated drug tolerance.

We also tested whether AST-120 terminates the effect of indole on induction of gene expression. Two sets of bacteria were grown with indole to the mid-logarithmic phase, at which point AST-120 was added to the first set of bacterial cultures, and its culture was continued to the stationary phase whereas the other set of bacterial cultures was similarly continued but without added AST-120. For this experiment, we chose the *acrD* promoter. We found that the level of LacZ expression in bacteria when AST-120 was added was lower than that of bacteria to which AST-120 was not added (Fig 2B).

## AST-120 decreases EspB level elevated by indole

EspB, which is secreted via the type III secretion system by EHEC, and this protein contributes to pathogenicity in host epithelial cells [31]. Previously, we found that indole increases secretion of this protein [14]. In this study, we tested the effect of AST-120 addition on EspB secretion. We grew wild-type parent EHEC and the *tnaA* mutant in Dulbecco's Modified Eagle Medium (DMEM) containing tryptophan or indole. In the preliminary experiment, we found that the wild-type parent produces 0.3 mM of indole in DMEM when tryptophan is added. We therefore used this concentration of indole for the EspB experiment (Fig 4A). We found that the addition of AST-120 reduced EspB levels in the wild-type parent (Fig 4B). As in our

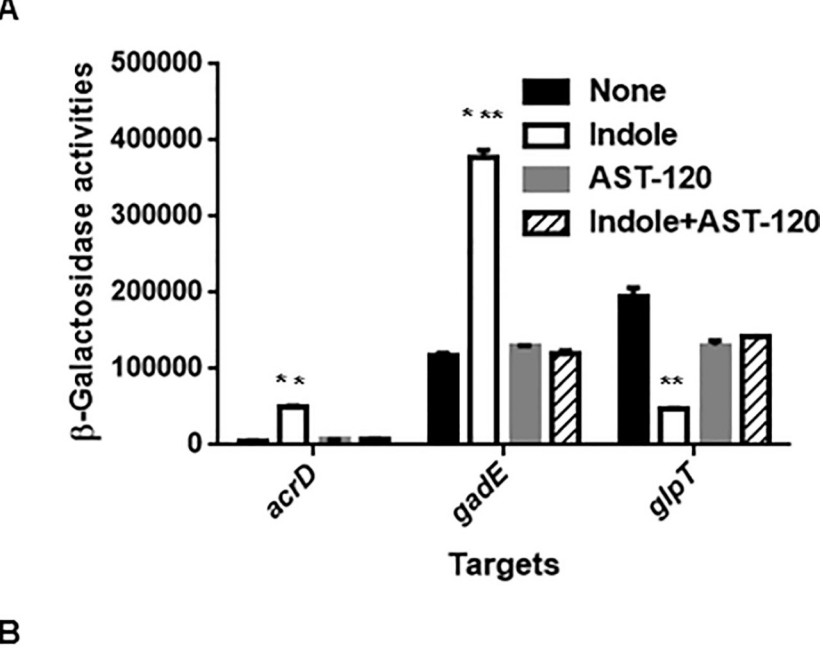

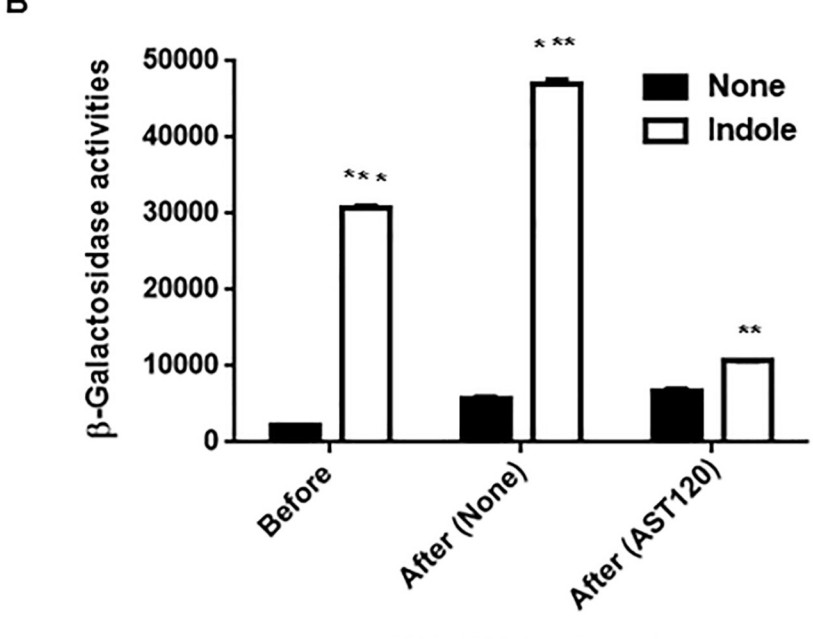

**Fig 2. Measurement of promoter activities.** (A) β-Galactosidase activities of MC4100 containing pNNacrD-P, pNNgadE-P or pNNglpT-P, the *lacZ* reporter plasmid grown with or without indole and/or AST-120. (B) β-Galactosidase activities correspond to *acrD* promoter activities in MC4100ΔtnaAB grown with or without indole before or after incubated with or without AST-120. β-Galactosidase activities represent values of chemiluminescence normalized by $OD_{600}$ correspond to the LacZ expression. Data plotted are the means from three independent experiments; error bars indicate the standard deviations, $**$, $P<0.01$; $***$, $P<0.001$. Asterisks denote significance for values of promoter activities in *E. coli* host cultured with indole relative to those in *E. coli* host cultured without indole.

previous study, EspB levels of the *tnaA* mutant when grown with tryptophan were lower than those of the wild-type parent because the *tnaA* mutant does not produce indole, and it was elevated when grown with exogenous indole. AST-120 addition decreased EspB levels in culture.

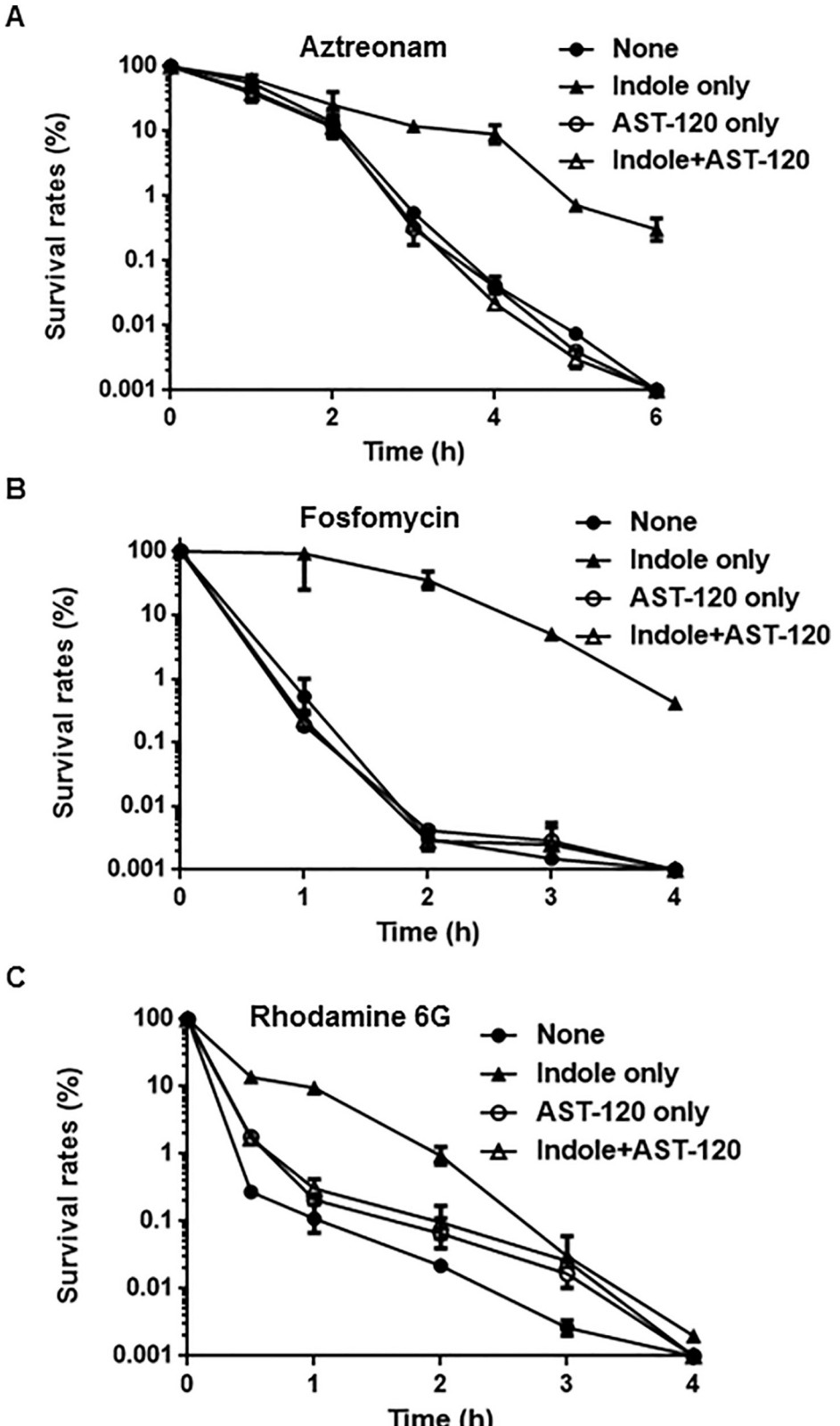

**Fig 3. Killing curves of MC4100 (for aztreonam and fosfomycin) and MC4100ΔacrB (for rhodamine 6G).** Bacteria were grown with or without indole and/or 30 mg AST-120, and thereafter incubated with 25 μg/ml aztreonam (A),

1.56 μg/ml fosfomycin (B) or 75 μg/ml rhodamine 6G (C). The survival rates at indicated time points were described as percent of CFUs for strains after incubation with the drug relative to those before incubation with the drug. Data plotted are the means and error bars indicate the ranges.

Thus, AST-120 reduces indole level, then the wild-type parent grown in the presence of AST-120 behaves as the *tnaA* mutant.

## AST-120 does not adsorb fosfomycin, rifampicin, colistin, or aminoglycosides, such as amikacin

AST-120 may adsorb not only indole and tryptophan but also some classes of antimicrobial agents. This property may limit the effectiveness of a combination therapy that uses AST-120 and a common antimicrobial agent. We tested the capability of AST-120 to adsorb antimicrobial agents that are generally active in *E. coli* species, including aztreonam and fosfomycin, to which tolerance is induced by indole. AST-120 adsorbed aztreonam, ciprofloxacin, minocycline, trimethoprim, and sulfamethoxazole, but not fosfomycin, rifampicin, colistin, or amikacin. Fosfomycin, rifampicin, colistin, and aminoglycosides, such as amikacin, are therefore still useful, even when AST-120 is present (Fig 5). In addition, AST-120 did not adsorb rhodamine 6G although it is not generally used for the treatment of bacterial infectious diseases.

## Discussion

AST-120 is used to treat progressive CKD by inhibiting the production of indoxyl sulfate, a uremic toxin derived from indole that is produced by some kinds of enteric bacteria, such as *E. coli* [21]. When kidneys function normally, the renal tubular cells remove indoxyl sulfate from the blood and eventually drain it with the urine [17]. However, in CKD patients, indoxyl-sulfate accumulates in the kidney and blood, where it generates reactive oxygen species (ROS), and induces oxidative cell damage [32]. Therefore, indoxyl sulfate not only accelerates kidney function decline but also damages vascular endothelial cells. Animal studies have shown that indoxyl sulfate disturbs the cardiovascular system, whereas administration of AST-120 alleviates it [33–34]. In this study, we propose that AST-120 plays another beneficial role against enteric infectious diseases caused by pathogenic *E. coli*, including EHEC, although this role may be limited because AST-120 adsorbs some antimicrobial agents. We provided evidence that AST-120 suppresses indole signaling of *E. coli* by adsorbing both indole and tryptophan (a substrate for indole production) and decreasing tolerance to particular antimicrobial agents, including aztreonam, fosfomycin, rhodamine 6G, and type III secretion proteins.

When bacteria communicate with other bacteria, they produce diffusible signaling molecules, such as autoinducers, and alter expression profiles in subsets of genes [5]. This activity is termed "bacterial cell-to-cell communication." There is increasing evidence that this activity is closely associated with drug resistance and pathogenicity and is therefore a possible efficient target for the treatment of refractory infectious diseases caused by pathogens resistant to traditional antibiotics. Gram-negative bacteria commonly use acyl-homoserine lactones (AHLs) as signaling molecules [8]. In addition to AHLs, some species also use autoinducer-2 [7]. *E. coli* uses indole instead of AHLs. It is produced not only by *E. coli* but also by other enteric bacteria, such as *Proteus* spp. [35]. Indole is synthesized from tryptophan, an essential amino acid contained in food, and some share of it is eventually converted into indoxyl sulfate [16]. However, a high level of indole can still accumulate at enteric sites [36]. Therefore, indole may increase resistance to some drugs and *E. coli* at enteric sites. AST-120 is expected to be able to remove indole. Indole also increases drug resistance of *Salmonella enterica via* induction of

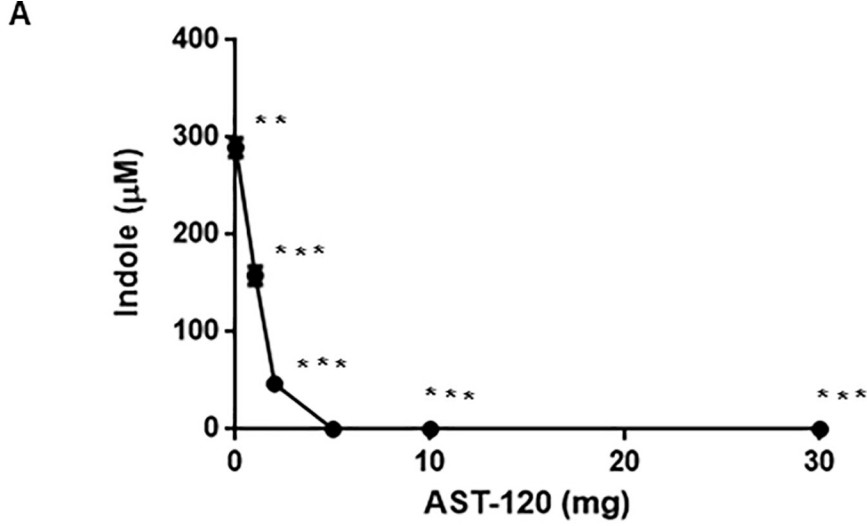

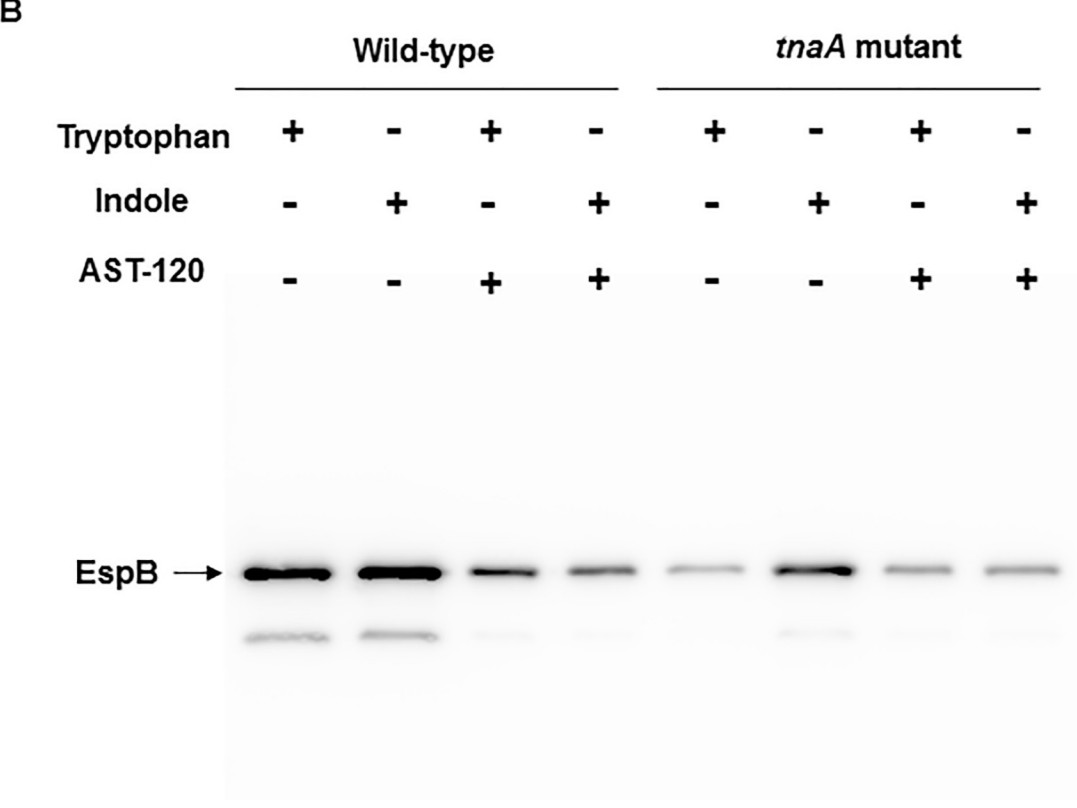

**Fig 4. Determination of indole and EspB levels in supernatant of EHEC O157.** (A) Indole concentration in culture supernatant of the wild-type parent EHEC O157 grown in DMEM containing 0.5 mM tryptophan with or without AST-120. Data plotted are the means from three independent experiments; error bars indicate the standard deviations, $^{**}$, $P < 0.01$; $^{***}$, $P < 0.001$. Asterisks denote significance for values of indole concentration in supernatant after incubated with AST-120 relative to those after incubated without AST-120. (B) The wild-type parent and the *tnaA* mutant were grown in DMEM containing 0.5 mM tryptophan or 0.3 mM indole with or without AST-120. Secreted proteins including EspB were separated by SDS/PAGE, and EspB was visualized by western blotting with EspB antiserum.

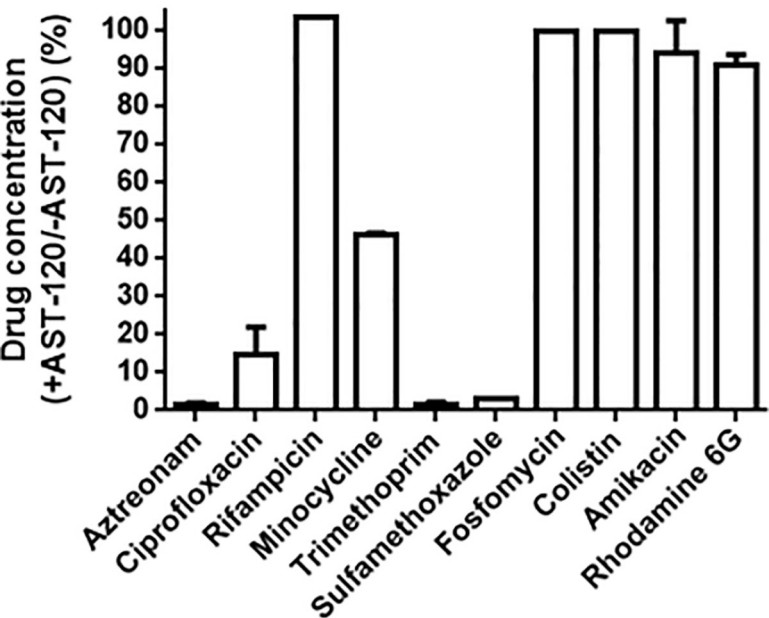

**Fig 5. Drug adsorption by AST-120.** Each drug in aqueous solution was incubated for 2 hours with or without AST-120. The *y* axis on the graph shows percent of drug concentration after incubation with AST-120 relative to drug concentration after incubation without AST-120. Data are the means of two biological replicates; error bars indicate the ranges.

AcrAB, a drug efflux pump, and biofilm formation of *Vibrio cholera* [37–38]. AST-120 administration may therefore contribute to therapies for these enteric infections. On the other hand, indole suppresses virulence of *P. aeruginosa* and *Staphylococcus aureus*, increases drug susceptibility in *Lysobacter* species and inhibits *P. aeruginosa* persister cell resuscitation [11–12, 39–40]. Therefore, we have to note that AST-120 may exhibit detrimental aspects in some of cases. However, we believe that AST-120 still has a merit for treatment of *E. coli* infections including EHEC.

Many studies report on the inhibition of bacterial signaling, and studies on AHL inhibitors are well known [6]. Although many cases involving antagonists target the receptors, some molecules inactivate signaling at the extracellular space without being incorporated into the bacterial cells. Cyclodextrins, for example, can capture AHLs in a matrix [41]. AST-120 inhibits indole signaling, and it works in extracellular spaces in a manner similar to that of cyclodextrins. These compounds may not be affected by membrane permeability and the presence of efflux pump. Some species of soil bacteria oxidize or degrade indole [42–43]. The enzymes catalyzing these reactions are also considered potential indole-signaling inhibitors. Compared with these degradation and modification enzymes, AST-120 is relatively small and stable.

However, tryptophan metabolites, including indole, also have beneficial aspects for host cells, therefore AST-120 may interfere with their beneficial actions. Studies in mice showed that low levels of indole alleviate liver and colonic inflammation in inflammatory bowel disease (IBD) [44–45]. Several *Clostridium* species, such as *C. sporogenes*, produce indole propionic acid (IPA) from tryptophan [46]. IPA promotes intestinal barrier function via activation of pregnane X receptor (PXR) while helping protect host cells from oxidative damage by scavenging hydroxyl radicals [47–48]. Tryptamine produced by *C. sporogenes* and *Ruminococcus gnavus* acts on enterochromaffin cells, inducing the release of 5-hydroxytryptamin, (5-HT) which stimulates gastrointestinal motility [49–50]. Treatment with indole-3-carboxaldehyde (ICA)

limits gut epithelial damage, reduces transepithelial bacterial translocation, and decreases inflammatory cytokine production [51]. Therefore, to enhance the potential efficacy of AST-120 therapy, establishing a method that minimizes interference of the beneficial effects caused by tryptophan metabolites may be important.

We note that AST-120 adsorbs some kinds of antimicrobial agents, such as aztreonam and ciprofloxacin which are commonly used to treat of *E. coli* infections. However, fosfomycin, colistin, rifampicin, and aminoglycosides are still not adsorbed by AST-120. Although fosfomycin is an old-generated antibiotic, it recently has attracted attention as a "last resort drug" to treat multidrug resistant (MDR) pathogens [52]. Increasingly, pathogenic *E. coli* clones that are resistant to quinolones and β-lactams, including carbapenem, have becomes a critical issue in clinical settings. Therefore, AST-120 may offer a potential benefit by helping fosfomycin therapy treat *E. coli* infections.

## Author Contributions

**Conceptualization:** Motoyuki Uchida, Fuyuhiko Nishijima.

**Data curation:** Hidetada Hirakawa, Haruyoshi Tomita.

**Formal analysis:** Hidetada Hirakawa, Haruyoshi Tomita.

**Funding acquisition:** Hidetada Hirakawa, Haruyoshi Tomita.

**Investigation:** Hidetada Hirakawa, Kumiko Kurabayashi, Ayako Takita.

**Methodology:** Kumiko Kurabayashi.

**Project administration:** Hidetada Hirakawa.

**Resources:** Motoyuki Uchida, Fuyuhiko Nishijima.

**Writing – original draft:** Hidetada Hirakawa.

**Writing – review & editing:** Hidetada Hirakawa, Motoyuki Uchida, Fuyuhiko Nishijima, Haruyoshi Tomita.

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
