## [Decision Letter · Decision Letter 0]

12 Feb 2020

PONE-D-20-01954

In vitro Activity of AST-120 That Suppresses Indole Signaling in Escherichia coli, Which Attenuates Drug Tolerance and Virulence

PLOS ONE

Dear Dr Hirakawa,

Thank you for submitting your manuscript to PLOS ONE. After careful consideration, we feel that it has merit but does not fully meet PLOS ONE’s publication criteria as it currently stands. Therefore, we invite you to submit a revised version of the manuscript that addresses the points raised during the review process.

The present methodology does not support the results and authors' conclusions. The authors should perform substantial changes to the methodology so as to address reviewers comments. 

h

We would appreciate receiving your revised manuscript by Mar 28 2020 11:59PM. To enhance the reproducibility of your results, we recommend that if applicable you deposit your laboratory protocols in protocols.io, where a protocol can be assigned its own identifier (DOI) such that it can be cited independently in the future. For instructions see: http://journals.plos.org/plosone/s/submission-guidelines#loc-laboratory-protocols

We look forward to receiving your revised manuscript.

Kind regards,

Amal Al-Bakri

Academic Editor

PLOS ONE

Journal Requirements:

We note that one or more of the authors are employed by a commercial company: Pharmaceuticals & Agrochemicals Division, Kureha Corporation.

4. Please ensure that you refer to Figures 4 and 5 in your text as, if accepted, production will need this reference to link the reader to the figure.

Reviewers' comments:

Reviewer's Responses to Questions

**Comments to the Author**

1. Is the manuscript technically sound, and do the data support the conclusions?

Reviewer #1: Yes

Reviewer #2: Partly

2. Has the statistical analysis been performed appropriately and rigorously? 

Reviewer #1: Yes

Reviewer #2: N/A

3. Have the authors made all data underlying the findings in their manuscript fully available?

Reviewer #1: Yes

Reviewer #2: No

4. Is the manuscript presented in an intelligible fashion and written in standard English?

Reviewer #1: Yes

Reviewer #2: Yes

5. Review Comments to the Author

Reviewer #1: This manuscript titled“In vitro Activity of AST-120 That Suppresses Indole Signaling in Escherichia coli, Which Attenuates Drug Tolerance and Virulence” describes a series of conclusions, and the experimental results are very detailed. However, the current status cannot be published yet, and the following modifications need to be completed before publication.

1. The introduction of indoles is not comprehensive enough, and there is a lack of multiple references, such as the recent studies by Thomas K Wood. Several new developments regarding Lysobacter were also not mentioned. These need to be added.

2. In Figure1, the significant change in statistics is only shown as P<0.01 and is only represented by single asterisk. The author needs to use one, two, and three asterisks for different P values.

3. In figure 1, the effect of 10mg ast-120 is obvious, what about 1mg or 5mg? This data needs to be shown in the diagram. This series of data needs to be improved.

4. English writing needs polishing.

5. There are many studies on the regulation of drug resistance by indoles. Previous studies have shown that indoles increase drug resistance in some bacteria and inhibit drug resistance in others. Are these background studies relevant to this study? Is there any inspiration? Need to be reflected in the discussion.

Reviewer #2: The manuscript describes a novel observation that AST-120 (Kremezin), used clinically for chronic kidney disease, is capable of adsorbing indole (in addition to a range of other molecules). One would naturally conclude that the addition of AST-120 to a bacterial culture will interfere with most indole-related signalling pathways, not because AST-120 is interfering in these pathways but simply because indole is absent/decreased. The authors demonstrate this expected outcome in a range of assays (antibiotic tolerance, regulation of genes encoding efflux pumps and fosfomycin permease & secretion of EspB protein). However, the authors claim that AST-120 interferes in these pathways. This claim is unsubstantiated as AST-120 has no effect on these pathways if indole is not present, as shown by their results. Therefore, these assays simply confirm the initial observation of AST-120 adsorbing indole without providing any added significance.

Abstract:

Lines 40-42: The results suggest only that AST-120 can adsorb a range of compounds. There is no data to support that a combination therapy of AST-120 with quinolones or B-lactams will lead to a better clinical outcome, hence the statement is unjustified.

Introduction:

Lines 73: The authors should explain the difference between resistance and tolerance for the benefit of the lay audience, I suggest referring to Nature Reviews Microbiology 14, 320 (2016) & mBio 10, e02095-19 (2019). Tolerant and persistent bacteria can survive antibiotic treatments without having acquired a resistance phenotype. In contrast to resistance, which enables bacteria to grow in a concentration of a drug that would otherwise prevent growth, tolerance is only a transient ability of bacteria to survive under otherwise bactericidal treatments.

Lines 75-81: The authors describe indole as a detrimental molecule on the host. This is inaccurate as indole has been reported to have both positive and negative effects (For example, see FEMS Nicrobiol Rev 34, 426-44 (2010) & Trends Microbiol 23, 707-718 (2015)).

Materials and Methods:

Lines 123-131: “Drug susceptibility assays” is described vaguely (what is the starting cfu/ml before treatment? Why incubation with no shaking? Why 60 min incubation for the antibiotics but 30 min for rhodamine G?). Also, the choice of the technique is odd.

As a starting point, MICs should be assayed for all tested antibiotics (and Rhodamine G) before and after the addition of AST-120 to confirm that we are looking at an increased tolerance not increased resistance. In contrast with resistance, tolerance do not lead to an increase in the MIC compared with susceptible bacteria (See Nature Reviews Microbiology 17, 441-448 (2019)).

Next, tolerance is characterised by a slower killing even at high concentrations of the drug, and that is assayed by either MDK99 (the minimum duration of treatment that kills 99% of the bacterial population) (See Nat Rev Microbiol 14:320–330 (2016)) or TDtest (see Sci Rep 7, 41284 (2017).

The authors need to comment on the suitability of their chosen method, considering that it is not how tolerance is usually assayed, unless the authors do not mean actually “tolerance”!

Lines 146-147: What is the basis of using OD280nm to calculate the concentrations of antibiotics? Why is it changes to 525nm for Rhodamine G?

Lines 148-149: The disk diffusion assays will be influenced by the presence of AST-120. I am wondering if the authors have tested whether AST-120 alone has any inhibitory effect on MG1655?

Results:

Line 170: What does “authentic” indole means?

Lines 172-175: The authors justify using 2 mM indole by describing that cultures of stationary phase E. coli in LB contain around 2 mM indole. This is buzzling as the concentration is known to be in the range of 0.3 – 0.7 mM, which is nowhere near 2 mM. See for example

- Di Martino, P., et al., Indole can act as extracellular signal to regulate biofilm formation of Escherichia coli and other indole-producing bacteria. Canadian journal of Microbilogy, 2003. 49(7): p. 443-449.

- Domka, J., J. Lee, and T.K. Wood, YliH (BssR) and YceP (BssS) Regulate Escherichia coli K-12 Biofilm Formation by Influencing Cell Signalling. Appl. Environ. Microbial., 2006. 72(4): p. 2449-2459.

- Hirakawa, H., et al., Secreted indole serves as a signal for expression of type III secretion system translators in enterohaemorrhagic Escherichia coli O157 : H7. Microbiology, 2009. 155(2): p. 541-550.

The authors need to comment on this contradiction.

Lines 177-178 & Fig 1B: The authors have shown than AST-120 adsorbs tryptophan as well as indole. Thus, when adding AST-120 to a tryptophan containing media, it is not possible to determine whether the reduction of indole is due to AST-120 adsorption of indole or tryptophan? This experiment is rather messy with no clear conclusion.

Lines 214-215: The authors conclude that “AST-120 suppresses AcrD-, MdtEF-, and GlpT-associated drug tolerance induced by indole”. This is not what the results show. The results simply show that AST-120 adsorbs indole (almost completely). This means that indole effects will be abolished by the addition of AST-120. Not because AST-120 interferes in the indole effects but simply because indole is not there anymore. In other words, AST-120 does NOT suppress AcrD-, MdtEF-, and GlpT-associated drug tolerance, but simply adsorbs indole so these effects are not there anymore.

Lines 216-222 & Fig 2B: Similar to my previous point, the addition of AST-120 did not affect the level of LacZ expression directly but simply adsorbed indole, thereby the expression of LacZ was decreased.

Lines 232-233: (Fig. 3A & B) should be (Fig. 4A & B).

Lines 237-238: Again, AST-120 does NOT decrease EspB levels as claimed by the authors. AST-120 adsorbs indole thus the WT strains behaves exactly as a ΔtnaA strain.

6. PLOS authors have the option to publish the peer review history of their article (what does this mean?). If published, this will include your full peer review and any attached files.

Reviewer #1: No

Reviewer #2: No

---

## [Author Response · Author response to Decision Letter 0]

26 Mar 2020

Thank you for reviewing our manuscript. According to your suggestions, we revised our manuscript. In addition, our response to your comments is described in our attached "Response to reviewers" file.

---

## [Decision Letter · Decision Letter 1]

16 Apr 2020

In vitro Activity of AST-120 That Suppresses Indole Signaling in Escherichia coli, Which Attenuates Drug Tolerance and Virulence

PONE-D-20-01954R1

Dear Dr. Hirakawa 

We are pleased to inform you that your manuscript has been judged scientifically suitable for publication and will be formally accepted for publication once it complies with all outstanding technical requirements.

With kind regards,

Amal Al-Bakri

Academic Editor

PLOS ONE

Additional Editor Comments (optional):

Reviewers' comments:

Reviewer's Responses to Questions

**Comments to the Author**

1. If the authors have adequately addressed your comments raised in a previous round of review and you feel that this manuscript is now acceptable for publication, you may indicate that here to bypass the “Comments to the Author” section, enter your conflict of interest statement in the “Confidential to Editor” section, and submit your "Accept" recommendation.

Reviewer #1: All comments have been addressed

2. Is the manuscript technically sound, and do the data support the conclusions?

Reviewer #1: Yes

3. Has the statistical analysis been performed appropriately and rigorously? 

Reviewer #1: Yes

4. Have the authors made all data underlying the findings in their manuscript fully available?

Reviewer #1: Yes

5. Is the manuscript presented in an intelligible fashion and written in standard English?

Reviewer #1: Yes

6. Review Comments to the Author

Reviewer #1: The author of this paper has answered the questions in a very complete way, and the logic and English writing have been improved.

7. PLOS authors have the option to publish the peer review history of their article (what does this mean?). If published, this will include your full peer review and any attached files.

Reviewer #1: No

---

## [Editor Report · Acceptance letter]

20 Apr 2020

PONE-D-20-01954R1 

*In vitro* activity of AST-120 that suppresses indole signaling in *Escherichia coli*, which attenuates drug tolerance and virulence 

Dear Dr. Hirakawa:

I am pleased to inform you that your manuscript has been deemed suitable for publication in PLOS ONE. Congratulations! Your manuscript is now with our production department. 

With kind regards,

on behalf of

Dr. Amal Al-Bakri 

Academic Editor

PLOS ONE